# SAFETY VERIFICATION OF MODEL BASED REINFORCEMENT LEARNING CONTROLLERS

## ABSTRACT

Model-based reinforcement learning (RL) has emerged as a promising tool for developing controllers for real world systems (e.g., robotics, autonomous driving, etc.). However, real systems often have constraints imposed on their state space which must be satisfied to ensure the safety of the system and its environment. Developing a verification tool for RL algorithms is challenging because the non-linear structure of neural networks impedes analytical verification of such models or controllers. To this end, we present a novel safety verification framework for model-based RL controllers using reachable set analysis. The proposed framework can efficiently handle models and controllers which are represented using neural networks. Additionally, if a controller fails to satisfy the safety constraints in general, the proposed framework can also be used to identify the subset of initial states from which the controller can be safely executed.

## 1 INTRODUCTION

One of the primary reasons for the growing application of reinforcement learning (RL) algorithms in developing optimal controllers is that RL does not assume *a priori* knowledge of the system dynamics. Model-based RL explicitly learns a model of the system dynamics, from observed samples of state transitions. This learnt model is used along with a planning algorithm to develop optimal controllers for different tasks. Thus, any uncertainties in the system, including environment noise, friction, air-drag etc., can also be captured by the modeled dynamics.

However, the performance of the controller is directly related to how accurately the learnt model represents the true system dynamics. Due to the discrepancy between the learnt model and the true model, the developed controller can behave unexpectedly when deployed on the real physical system, e.g., land robots, UAVs, etc. (Benbrahim & Franklin, 1997; Endo et al., 2008; Morimoto & Doya, 2001). This unexpected behavior may result in the violation of constraints imposed on the system, thereby violating its safety requirements (Moldovan & Abbeel, 2012). Thus, it is necessary to have a framework which can ensure that the controller will satisfy the safety constraints *before* it is deployed on a real system. This raises the primary question of interest: *Given a set of safety constraints imposed on the state space, how do we determine whether a given controller is safe or not?*

In the literature, there have been several works that focus on the problem of ensuring safety. Most of these works incorporate safety constraints in the learning phase to train a controller (policy) to satisfy certain desired specifications or constraints. However, to achieve this goal, some works make strict assumptions on the complete or accurate knowledge of the system dynamics (Zheng & Ratliff, 2020; Hasanbeig et al., 2020) which can be difficult to obtain. Further, to incorporate safety during learning, some works approximate the original problem to represent safety constraints in a tractable form (Fu et al., 2018; Avni et al., 2019), which reduces the performance of the final trained controller (Fu et al., 2018; Eriksson & Dimitrakakis, 2019; Junges et al., 2016; Könighofer et al., 2020). On the other hand, some of the works aim at *finding* a safe controller, under the assumption of a known baseline safe policy (Hans et al., 2008; Garcia & Fernández, 2012; Berkenkamp et al., 2017; Thomas et al., 2015; Laroche et al., 2019; Zheng & Ratliff, 2020), or several known safe policies (Perkins & Barto, 2002). However, such safe policies may not be readily available in general. Alternatively, Akametalu et al. (2014) used reachability analysis to develop safe model-based controllers, under the

assumption that the system dynamics can be modeled using Gaussian processes, i.e., an assumption which is violated by most modern RL methods that make use of neural networks (NN) instead.

While there have been several works proposed to develop safe controllers, some of the assumptions made in these works may not be possible to realize in practice. In recent years, this limitation has drawn attention towards developing *verification* frameworks for RL controllers, which is the focus of this paper. The *safety verification* algorithm proposed in this work is a standalone framework which makes no assumptions on how the model-based RL controller is trained. It works independently of the training phase to identify the safe initial conditions for any given policy. One advantage of using a standalone verification framework is that we can deploy potentially unsafe policies on real systems, without further training, by restricting their initial conditions to only the safe states. Since verifying safety of an NN based RL controller is also related to verifying the safety of the underlying NN model (Xiang et al., 2018b; Tran et al., 2019b; Xiang et al., 2018a; Tran et al., 2019a), we provide an additional review for these methods in Appendix A.1.

**Contributions:** In this work, we focus on the problem of determining whether a given controller is safe or not, with respect to satisfying constraints imposed on the state space. To do so, we propose a novel safety verification algorithm for model-based RL controllers using *forward reachable tube* analysis that can handle NN based learnt dynamics and controllers, while also being robust against modeling error. The problem of determining the reachable tube is framed as an optimal control problem using the Hamilton Jacobi (HJ) partial differential equation (PDE), whose solution is computed using the level set method. The advantage of using the level set method is the fact that it can represent sets with non-convex boundaries, thereby avoiding approximation errors that most existing methods suffer from. Additionally, if a controller is deemed unsafe, we take a step further to identify if there are any starting conditions from which the given controller can be safely executed. To achieve this, a *backward reachable tube* is computed for the learnt model and, to the best of our knowledge, this is the first work which computes the backward reachable tube over an NN. Finally, empirical results are presented on two domains inspired by real-world applications where safety verification is critical.

## 2 PROBLEM SETTING

Let $\mathbb{S} \subset \mathbb{R}^n$ denote the set of states and $\mathbb{A} \subset \mathbb{R}^m$ denote the set of feasible actions for the RL agent. Let $\mathbb{S}_0 \subset \mathbb{S}$ denote the set of bounded initial states and $\xi := \{(\boldsymbol{s}_t, \boldsymbol{a}_t)\}_{t=0}^T$ represent a trajectory generated over a finite time $T$, as a sequence of state and action tuples, where subscript $t$ denotes the instantaneous time. Additionally, let $\boldsymbol{s}(\cdot)$ and $\boldsymbol{a}(\cdot)$ represent a sequence of states and actions, respectively. The state constraints imposed on the system are represented as unsafe regions using bounded sets $\mathbb{C}_s = \cup_{i=1}^p \mathbb{C}_s^{(i)}$, where $\mathbb{C}_s^{(i)} \subset \mathbb{S}$, $\forall i \in \{1, 2, \dots p\}$. The true system dynamics is given by a non-linear function $f : \mathbb{S} \times \mathbb{A} \to \mathbb{R}^n$ such that, $\dot{\boldsymbol{s}} = f(\boldsymbol{s}, \boldsymbol{a})$, and is unknown to the agent.

A model-based RL algorithm is used to find an optimal controller $\pi : \mathbb{S} \to \mathbb{A}$, to reach a set of target states $\mathbb{T} \subset \mathbb{S}$ within some finite time $T$, while avoiding constraints $\mathbb{C}_s$. An NN model, $\hat{f}_{\boldsymbol{\theta}} : \mathbb{S} \times \mathbb{A} \to \mathbb{R}^n$ parameterized by weights $\boldsymbol{\theta}$, is trained to learn the true, but unknown, system dynamics from the observed state transition data tuples $D = \{(\boldsymbol{s}_t, \boldsymbol{a}_t, \Delta \boldsymbol{s}_{t+1})^{(i)}\}_{i=1}^N$. However, due to sampling bias, the learnt model $\hat{f}_{\boldsymbol{\theta}}$ may not be accurate. We assume that it is possible to estimate a bounded set $\mathbb{D} \subset \mathbb{R}^n$ such that, at any state $\boldsymbol{s} \in \mathbb{S}$, augmenting the learnt dynamics $\hat{f}_{\boldsymbol{\theta}}$ with some $\boldsymbol{d} \in \mathbb{D}$ results in a closer approximation of the true system dynamics at that particular state. Using this notation, we now define the problem of safety verification of a given controller $\pi(\boldsymbol{s})$.

**Problem 1** *(Safety verification): Given a set of initial states $\mathbb{S}_0$, determine if $\forall \boldsymbol{s}_0 \in \mathbb{S}_0$, all the trajectories $\xi$ executed under $\pi(\boldsymbol{s})$ and following the system dynamics $f$, satisfy the constraints $\mathbb{C}_s$ or not.*

The solution to Problem 1 will only provide a binary yes or no answer to whether $\pi(\boldsymbol{s})$ is safe or not with respect to $\mathbb{S}_0$. In the case where the policy is unsafe, a stronger result is the identification of safe initial states $\mathbb{S}_{safe} \subset \mathbb{S}_0$ from which $\pi(\boldsymbol{s})$ executes trajectories which always satisfy the constraints $\mathbb{C}_s$. This problem is stated below.

**Problem 2** *(Safe initial states): Given $\pi(\boldsymbol{s})$, find $\mathbb{S}_{safe}$, such that, any trajectory $\xi$ executed under $\pi(\boldsymbol{s})$ and following the system dynamics $f$, starting from any $\boldsymbol{s}_0 \in \mathbb{S}_{safe}$, satisfies the constraints $\mathbb{C}_s$.*

## 3    SAFETY VERIFICATION

To address Problems 1 and 2, we use *reachability analysis*. Reachability analysis is an exhaustive verification technique which tracks the evolution of the system states over a finite time, known as the *reachable tube*, from a given set of initial states $\mathbb{S}_0$ (Maler, 2008). If the evolution is tracked starting from $\mathbb{S}_0$, then it is called the *forward reachable tube* and is denoted as $\mathbb{F}_R(T)$. Analogously, if the evolution is tracked starting from $\mathbb{C}_s$ to $\mathbb{S}_0$, then it is called the *backward reachable tube* and is denoted as $\mathbb{B}_R(T)$.

In the following sections, we will formulate a reachable tube problem for NN-based models and controllers, and then propose a verification framework that (a) can determine whether or not a given policy $\pi$ is safe, and (b) can compute $\mathbb{S}_{safe}$ if $\pi$ is unsafe. To do so, there are two main questions that need to be answered. First, since the true system dynamics $f$ is unknown, how can we determine a conservative bound on the modeling error, to augment the learnt model $\hat{f}_{\boldsymbol{\theta}}$ and better model the true system dynamics when evaluating a controller $\pi$? Second, how do we formulate the *forward* and *backward reachable tube* problems over the NN modeled system dynamics?

### 3.1    MODEL-BASED REINFORCEMENT LEARNING

In this section, we focus on the necessary requirements for the modeled dynamics $\hat{f}_{\boldsymbol{\theta}}$ and discuss the estimation of the modeling error set $\mathbb{D}$. A summary of the model-based RL framework is presented in Appendix A.2. Recall that the learnt model $\hat{f}_{\boldsymbol{\theta}}$ is represented using an NN and predicts the change in states $\Delta \hat{\boldsymbol{s}}_{t+1} \in \mathbb{R}^n$. To learn $\hat{f}_{\boldsymbol{\theta}}$, an observed data set $D = \{(\boldsymbol{s}_t, \boldsymbol{a}_t, \Delta \boldsymbol{s}_{t+1})^{(i)}\}_{i=1}^N$ is first split into a training data set $D_t$ and a validation data set $D_v$. A supervised learning technique is then used to train $\hat{f}_{\boldsymbol{\theta}}$ over $D_t$ by minimizing the prediction error $E = \frac{1}{N_t} \sum_{i=1}^{N_t} ||\Delta \hat{\boldsymbol{s}}_{t+1}^{(i)} - \Delta \boldsymbol{s}_{t+1}^{(i)}||^2$, where $\Delta \hat{\boldsymbol{s}}_{t+1}$ is the change predicted by the learnt model, $\Delta \boldsymbol{s}_{t+1}$ is the true observed change in state and $N_t = |D_t|$. With this notation, we now formalize the following necessary assumption for this work. This assumption is required to ensure boundedness during analysis and is easily satisfied by NNs that use common activation functions like `tanh` or `sigmoid` (Usama & Chang, 2018).

**Assumption 1**   $\hat{f}_{\boldsymbol{\theta}}$ *is Lipschitz continuous and* $\forall j \in \{1, ..., n\}, \Delta \hat{\boldsymbol{s}}_j \in [-c, c]$, *where* $|c| < \infty$.

**Modeling error:** As mentioned earlier, the accuracy of the learnt model $\hat{f}_{\boldsymbol{\theta}}$ depends on the quality of data and the NN being used, thereby resulting in some modeling error $\boldsymbol{d}$ in the prediction of the next state. Estimating modeling errors is an active area of research and is required for several existing works on safe RL (Akametalu et al., 2014; Gillula & Tomlin, 2012), and is complementary to our goal. Since the primary contribution of this work is the development of a reachable tube formulation for model-based controllers that use NNs, we rely on existing techniques (Moldovan et al., 2015) to estimate a conservative modeling error bound. We leverage the error estimates $\hat{\boldsymbol{d}}_j = \hat{\Delta} s_{t+1}^{(j)} - \Delta s_{t+1}^{(j)}$, of $\hat{f}_{\boldsymbol{\theta}}$, for the transition tuples in the validation set $D_v$ to construct the upper confidence bound $\boldsymbol{d}^+ = [d_1, \ d_2, \ldots d_n]$ and lower confidence bound $\boldsymbol{d}^- = [-d_1, -d_2, \ldots -d_n]$ for $\boldsymbol{d}$, for each state dimension. Let a high-confidence bounded error set $\mathbb{D}$ be defined such that $\mathbb{D} = \{\boldsymbol{d} : \forall i \in \{1, ..., n\}, \ \boldsymbol{d}_i^- < \boldsymbol{d}_i < \boldsymbol{d}_i^+\}$. We then use $\mathbb{D}$ to represent the augmented learnt system dynamics as

$$\hat{f}_{\boldsymbol{\theta}}^{(r)} := \hat{f}_{\boldsymbol{\theta}}(\boldsymbol{s}, \boldsymbol{a}) + \boldsymbol{d}, \ \ \boldsymbol{d} \in \mathbb{D}. \tag{1}$$

### 3.2    REACHABLE TUBE FORMULATION

For an exhaustive verification technique on a continuous state and action space, it is infeasible to sample trajectories from *every* point in the given initial state and further, to verify whether *all* these trajectories satisfy the safety constraints. Therefore, reachable sets are usually (approximately) represented as convex polyhedrons and their evolution is tracked by pushing the boundaries of this polyhedron according to the system dynamics. However, as convex polyhedrons can lead to large

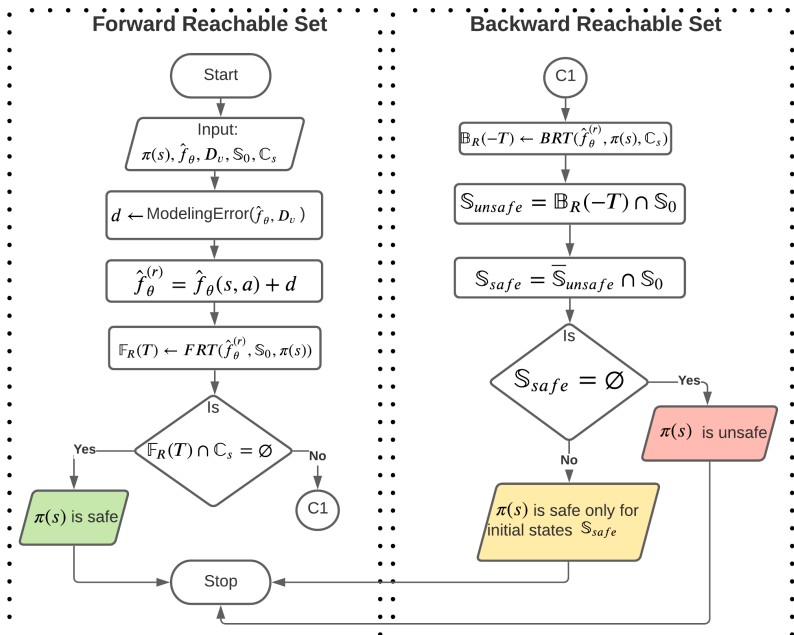

Figure 1: Flowchart of the proposed safety verification algorithm. The rectangle on the left represents the flow for computing the forward reachable tube (FRT), which can only state if $\pi(s)$ is safe or unsafe for $\mathbb{S}_0$. The rectangle on the right presents the flow for computing the backward reachable tube (BRT), which is invoked if $\pi(s)$ is unsafe. The BRT analysis can compute the subset of safe initial states $\mathbb{S}_{safe}$ for $\pi(s)$, if such a set exists.

approximation errors, we leverage the level set method (Mitchell et al., 2005) to compute the boundaries of the reachable tube for NN-based models and controllers at every time instant. With the level set method, even non-convex boundaries of the reachable tube can be represented, thereby ensuring an accurate computation of the reachable tube (Mitchell et al., 2005). Since the non-linear, non-convex structure of NNs is not suitable for analytical verification techniques, the reachability analysis gives an efficient, simulation-based approach to analyze the safety of the controller. Therefore, in this subsection, we formulate the reachable tube for a NN modeled system dynamics $\hat{f}_{\boldsymbol{\theta}}$, but first we formally define the forward reachable tube and backward reachable tube for a policy $\pi(s)$.

**Forward reachable tube (FRT):** It is the set of all states that can be *reached from* an initial set $\mathbb{S}_0$, when the trajectories $\xi$ are executed under policy $\pi(s)$ and system dynamics $\hat{f}_{\boldsymbol{\theta}}^{(r)}(\boldsymbol{s}, \boldsymbol{a}, \boldsymbol{d})$. The FRT is computed over a finite length of time $T$ and is formally defined as

$$\mathbb{F}_R(T) := \{ \boldsymbol{s} : \forall \boldsymbol{d} \in \mathbb{D}, \boldsymbol{s}(\cdot) \text{ satisfies } \dot{\boldsymbol{s}} = \hat{f}_{\boldsymbol{\theta}}^{(r)}(\boldsymbol{s}, \boldsymbol{a}, \boldsymbol{d}), \text{ where } \boldsymbol{a} = \pi(\boldsymbol{s}), \ \boldsymbol{s}_{t_0} \in \mathbb{S}_0, t_f = T \}, \quad (2)$$

where $t_0$ and $t_f$ denote the initial and final time of the trajectory $\xi$, respectively.

**Backward reachable tube (BRT):** It is the set of all states which *can reach* a given bounded target set $\mathbb{T} \subset \mathbb{R}^n$, when the trajectories $\xi$ are executed under policy $\pi(s)$ and system dynamics $\hat{f}_{\boldsymbol{\theta}}^{(r)}(\boldsymbol{s}, \boldsymbol{a}, \boldsymbol{d})$. The BRT is also computed for a finite length of time, with the trajectories starting at time $t_0 = -T$ and ending at time $t_f = 0$. It is denoted as

$$\mathbb{B}_R(-T) := \{ \boldsymbol{s}_0 : \forall \boldsymbol{d} \in \mathbb{D}, \boldsymbol{s}(\cdot) \text{ satisfies } \dot{\boldsymbol{s}} = \hat{f}_{\boldsymbol{\theta}}^{(r)}(\boldsymbol{s}, \boldsymbol{a}, \boldsymbol{d}), \text{ where } \boldsymbol{a} = \pi(\boldsymbol{s})$$
$$\text{with } \boldsymbol{s}_{t_0} = \boldsymbol{s}_{-T}; \ \boldsymbol{s}_{t_f} \in \mathbb{T}, t_f \in [-T, 0] \}. \quad (3)$$

The key difference between the FRT and BRT is that, for the former, the initial set of states are known, whereas for the latter, the final set of states are known.

**Outline:** The flowchart of the safety verification framework proposed in this work is presented in Fig. 1. Given a model-based policy $\pi(s)$, the set of initial states $\mathbb{S}_0$ and the set of constrained states $\mathbb{C}_s$, the first step is to estimate the bounded set of modeling error $\mathbb{D}$, as discussed in Section 3.1. Using $\hat{f}_{\boldsymbol{\theta}}^{(r)}$, the FRT is constructed from the initial set $\mathbb{S}_0$ and it contains all the states reachable by $\pi(s)$ over a finite time $T$. Thus, if the FRT contains any state from the unsafe region $\mathbb{C}_s$, $\pi(s)$ is deemed *unsafe*. Therefore, the solution to Problem 1, is determined by analyzing the set of intersection of the FRT with $\mathbb{C}_s$ as

$$\pi = \begin{cases} \text{safe} & \text{if } \mathbb{F}_R(T) \cap \mathbb{C}_s = \emptyset, \\ \text{unsafe} & \text{if } \mathbb{F}_R(T) \cap \mathbb{C}_s \neq \emptyset. \end{cases} \tag{4}$$

If $\pi(s)$ is classified as *safe* for the entire set $\mathbb{S}_0$, then no further analysis is required. However, if $\pi(s)$ is classified as *unsafe*, we proceed to compute the subset $\mathbb{S}_{\text{safe}} \subset \mathbb{S}_0$ of initial states for which $\pi(s)$ generates safe trajectories. $\mathbb{S}_{safe}$ is the solution to Problem 2 and allows an unsafe policy to be deployed on a real system, with restrictions on the starting states. To this end, the BRT is computed from the unsafe region $\mathbb{C}_s$, to determine the set of trajectories (and states) which terminate in $\mathbb{C}_s$. The intersection of the BRT with $\mathbb{S}_0$ determines the set of unsafe initial states $\mathbb{S}_{unsafe}$. To determine $\mathbb{S}_{safe}$, we utilize the following properties, (a) $\mathbb{S}_{safe} \cup \mathbb{S}_{unsafe} = \mathbb{S}_0$, and (b) $\mathbb{S}_{safe} \cap \mathbb{S}_{unsafe} = \emptyset$, and compute

$$\mathbb{S}_{safe} = \overline{\mathbb{S}}_{unsafe} \cap \mathbb{S}_0. \tag{5}$$

If $\mathbb{S}_{safe} \neq \emptyset$, then we have identified the safe initial states for $\pi(s)$, otherwise, it is concluded that there are no initial states in $\mathbb{S}_0$ from which $\pi(s)$ can generate safe trajectories.

**Mathematical Formulation:** This section presents the mathematical formulation to compute the BRT. The FRT can be computed with a slight modification to the BRT formulation and this is discussed in the end of this section.

Recall, for the BRT problem, there exists a target set $\mathbb{T} \subset \mathbb{R}^n$ which the agent has to reach in finite time, i.e., the condition on the final state is given as $s_{t_f} \in \mathbb{T}$. Conventionally, for the BRT formulation, the final time $t_f = 0$ and the starting time $t_0 = -T$, where $0 < T < \infty$. When evaluating a policy $\pi(s)$, the controller input is computed by the given policy as $\boldsymbol{a} = \pi(s)$. However, following the system dynamics $\hat{f}_{\boldsymbol{\theta}}^{(r)}$ in (1), the modeling error $\boldsymbol{d}$ is now included in the system as an adversarial input, whose value at each state is determined so as to maximize the controller's cost function. We use the HJ PDE to formulate the effect of the modeling error on the system for computing the BRT, but first we briefly review the formulation of the HJ PDE with an NN modeled system dynamics $\hat{f}_{\boldsymbol{\theta}}$ in the following.

For an optimal controller, we first define the cost function which the controller has to minimize. Let $C(\boldsymbol{s}_t, \boldsymbol{a}_t)$ denote the running cost of the agent, which is dependent on the state and action taken at time $t \in [-T, 0]$. Let $g(\boldsymbol{s}_{t_f})$ denote the cost at the final state $\boldsymbol{s}_{t_f}$. Then, the goal of the optimal controller is to find a series of optimal actions such that

$$\min_{\boldsymbol{a}_\tau(\cdot)} \left( \int_{-T}^0 C(\boldsymbol{s}_\tau, \boldsymbol{a}_\tau) d\tau + g(\boldsymbol{s}_{t_f}) \right) \tag{6}$$

$$\text{subject to } \dot{\boldsymbol{s}} = \hat{f}_{\boldsymbol{\theta}}(\boldsymbol{s}, \boldsymbol{a}), \quad \boldsymbol{s}_{t_f} \in \mathbb{T},$$

where $\boldsymbol{a}_\tau \in \mathbb{A}$ and $\hat{f}_{\boldsymbol{\theta}}$ is the NN modeled system dynamics. The above optimization problem is solved using the *dynamic programming* approach (Smith & Smith, 1991), which is based on the *Principle of Optimality* (Troutman, 2012). Let $V(\boldsymbol{s}_t, t)$ denote the value function of a state $s$ at time $t \in [-T, 0]$, such that

$$V(\boldsymbol{s}_t, t) = \min_{\boldsymbol{a}_\tau(\cdot)} \left[ \int_t^0 C(\boldsymbol{s}_\tau, \boldsymbol{a}_\tau) d\tau + g(\boldsymbol{s}_{t_f}) \right] = \min_{\boldsymbol{a}_\tau(\cdot)} \left[ \int_t^{t+\delta} C(\boldsymbol{s}_\tau, \boldsymbol{a}_\tau) d\tau + V(\boldsymbol{s}_{t+\delta}, t+\delta) \right], \tag{7}$$

where $\delta > 0$. $V(\boldsymbol{s}_t, t)$ is a quantitative measure of being at a state $s$, described in terms of the cost required to reach the goal state from $s$. Then, using the Taylor series expansion, $V(\boldsymbol{s}_{t+\delta}, t+\delta)$ is approximated around $V(\boldsymbol{s}_t, t)$ in (7) to derive the HJ PDE as

$$\frac{dV}{dt} + \min_{\boldsymbol{a}} \left[ \nabla V \cdot \hat{f}_{\boldsymbol{\theta}}(\boldsymbol{s}, \boldsymbol{a}) + C(\boldsymbol{s}, \boldsymbol{a}) \right] = 0, \tag{8}$$

$$V(\boldsymbol{s}_{t_f}, t_f) = g(\boldsymbol{s}_{t_f}),$$

where $\nabla V \in \mathbb{R}^n$ is the spatial derivative of $V$. Additionally, the time index has been dropped above and the dynamics constraint in (6) has been included in the PDE. Equation (8) is a terminal value PDE, and by solving (8), we can compute the value of a state $V(\boldsymbol{s}_t, t)$ at any time $t$.

We now discuss how the formulation in (8) can be modified to obtain the BRT. It is noted that along with computing the value function, the formulation in (8) also computes the optimal action $\boldsymbol{a}$. However, in Problems 1 and 2, the optimal policy $\pi(\boldsymbol{s})$ is already provided. Therefore, the constraint $\boldsymbol{a} = \pi(\boldsymbol{s})$ should be included in problem (6), thereby avoiding the need of minimizing over actions $\boldsymbol{a} \in \mathbb{A}$ in (8). Additionally, as discussed in Section 3.1, the NN modeled system dynamics $\hat{f}_{\boldsymbol{\theta}}$ may not be a good approximation of the true system dynamics $f$. Instead, the augmented learnt system dynamics $\hat{f}_{\boldsymbol{\theta}}^{(r)}$ in (1) is used in place of $\hat{f}_{\boldsymbol{\theta}}$ in (8), since it better models the true dynamics at a given state. However, by including $\hat{f}_{\boldsymbol{\theta}}^{(r)}$ in (8), the modeling error $\boldsymbol{d}$ is now included in the formulation. The modeling error $\boldsymbol{d} \in \mathbb{D}$ is treated as an adversarial input which is trying to drive the system away from it's goal state by taking a value which maximizes the cost function at each state. Thus, to account for this adversarial input, the formulation in (8) is now maximized over $\boldsymbol{d}$.

Lastly, the BRT problem is posed for a set of states and not an individual state. Hence, an efficient representation of the target set is required to propagate an entire set of trajectories at a time, as opposed to propagating individual trajectories.

**Assumption 2** *The target set* $\mathbb{T} \subset \mathbb{R}^n$ *is closed and can be represented as the zero sublevel set of a bounded and Lipschitz continuous function* $l : \mathbb{R}^n \to \mathbb{R}$*, such that,* $\mathbb{T} = \{\boldsymbol{s} : l(\boldsymbol{s}) \leq 0\}$*.*

The above assumption defines a function $l$ to check whether a state lies inside or outside the target set. If $\mathbb{T}$ is represented using a regular, well-defined geometric shape (like a sphere, rectangle, cylinder, etc.), then deriving the level set function $l(\boldsymbol{s})$ is straight forward, whereas, an irregularly shaped $\mathbb{T}$ can be represented as a union of several well-defined geometric shapes to derive $l(\boldsymbol{s})$.

For the BRT problem, the goal is to determine all the states which can reach $\mathbb{T}$ within a finite time. The path taken by the controller is irrelevant and only the value of the final state is used to determine if any state $\boldsymbol{s} \in \mathbb{B}_R(-T)$. From Assumption 2, the terminal condition $\boldsymbol{s}_{t_f} \in \mathbb{T}$ can be restated as $l(\boldsymbol{s}_{t_f}) \leq 0$. Thus, to prevent the system from reaching $\mathbb{T}$, the adversarial input $\boldsymbol{d}$ tries to maximize $l(\boldsymbol{s}_{t_f})$, thereby pushing $\boldsymbol{s}_{t_f}$ as far away from $\mathbb{T}$ as possible. Therefore, the cost function in (6) is modified to $J = l(\boldsymbol{s}_{t_f})$. Additionally, any state which can reach $\mathbb{T}$ *within* a finite time interval $T$ is included in the BRT. Therefore, if any trajectory reaches $\mathbb{T}$ at some $t_f < 0$, it shouldn't be allowed to leave the set. Keeping this in mind, the BRT optimization problem can be posed as

$$\max_{\boldsymbol{d}(\cdot)} \left( \min_{t \in [-T, 0]} l(\boldsymbol{s}_{t_f}) \right) \tag{9}$$
$$\text{subject to: } \dot{\boldsymbol{s}} = \hat{f}_{\boldsymbol{\theta}}^{(r)}(\boldsymbol{s}, \boldsymbol{a}, \boldsymbol{d}), \quad \boldsymbol{a} = \pi(\boldsymbol{s}), \quad l(\boldsymbol{s}_{t_f}) \leq 0,$$

where the inner minimization over time prevents the trajectory from leaving the target set. Then, the value function for the above problem is defined as

$$V_R(\boldsymbol{s}_t, t) := \max_{\boldsymbol{d}(\cdot)} l(\boldsymbol{s}(t_f)). \tag{10}$$

Comparing this with (7), it is observed that the value of a state $\boldsymbol{s}$ is no longer dependent on the running cost $C(\boldsymbol{s}, \boldsymbol{a})$. This doesn't imply that the generated trajectories are not optimal w.r.t. action $\boldsymbol{a}$, because the running cost is equivalent to the negative reward function, for which $\pi(\boldsymbol{s})$ is already optimized. Instead, $V_R$ solely depends on whether the final state of the trajectory lies within the target set or not, i.e., whether or not $\boldsymbol{s}_{t_f} \in \mathbb{T}$. Thus, the value function $V_R$ for any state $\boldsymbol{s}$ is equal to $l(\boldsymbol{s}_{t_f})$, where $\boldsymbol{s}_{t_f}$ is the final state of the trajectory originating at $\boldsymbol{s}$. Then, the HJ PDE for the problem in (9) is stated as

$$\frac{dV_R}{dt} + min\{0, H^*(\boldsymbol{s}, \nabla V_R(\boldsymbol{s}_t, t), t)\} = 0,$$
$$V_R(\boldsymbol{s}_{t_f}, t_f) = l(\boldsymbol{s}_{t_f}), \tag{11}$$
$$\text{where } H^* = \max_{\boldsymbol{d}} \left( \nabla V_R \cdot \hat{f}_{\boldsymbol{\theta}}^{(r)}(\boldsymbol{s}, \pi(\boldsymbol{s}), \boldsymbol{d}) \right),$$

where $H^*$ represents the optimal Hamiltonian. Since we are computing the BRT, $min\{0, H^*(\boldsymbol{s}, \nabla V_R(\boldsymbol{s}_t, t), t)\}$ in the PDE above ensures that the tube grows only in the backward

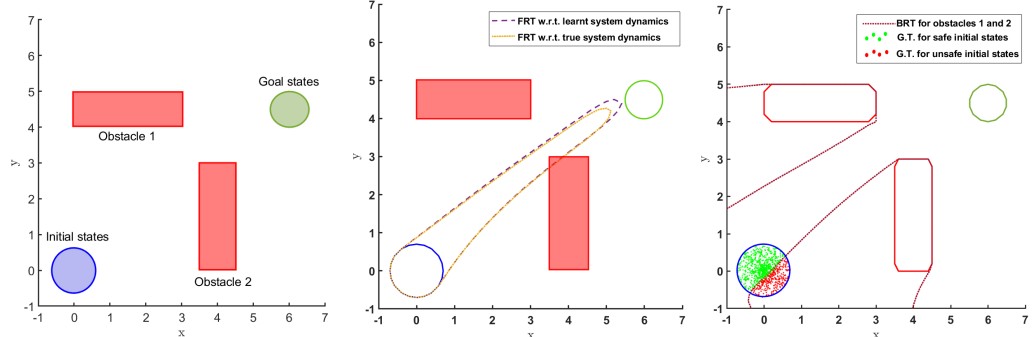

Figure 2: (Left) The environment for the safe land navigation problem. (Middle) The FRTs for both the augmented learnt model dynamics and true system dynamics classify the controller as unsafe. (Right) The BRT computed from obstacles 1 and 2 for the given controller. $\mathbb{S}_{safe}$ computed by the proposed BRT algorithm is compared with the ground truth (G.T.) data for the safe initial states, marked in green.

direction, thereby preventing a trajectory which has reached $\mathbb{T}$ from leaving. In Problem (11), the optimal action has been substituted by $\pi(s)$ and the augmented learnt dynamics is used instead of $\hat{f}_\theta$. The Hamiltonian optimization problem can be further simplified to derive an analytical solution for the modeling error. By substituting the augmented dynamics from (1), the optimization problem can be re-written as

$$H^* = \nabla V_R \cdot \hat{f}_\theta(s, a) + \max_d \nabla V_R \cdot d. \tag{12}$$

Expanding $\nabla V_R = [p_1, p_2, \ldots, p_n]^T \in \mathbb{R}^n$, the vector product $\nabla V_R \cdot d = p_1 d_1 + p_2 d_2 + \ldots + p_n d_n$. Therefore, to maximize $\nabla V_R \cdot d$, the disturbance control is chosen as

$$d_i = \begin{cases} d_i & \text{if } p_i > 0 \\ -d_i & \text{if } p_i < 0 \end{cases}, \forall i = 1, \ldots n. \tag{13}$$

With this analytical solution, the final PDE representing the BRT is stated as

$$\frac{dV_R}{dt} + min\{0, H^*(s, \nabla V_R(s_t, t), t)\} = 0,$$
$$V_R(s_{t_f}, t_f) = l(s_{t_f}), \tag{14}$$
$$\text{where } H^* = \nabla V_R \cdot \hat{f}_\theta(s, a) + (|p_1|d_1 + |p_2|d_2 + \ldots + |p_n|d_n).$$

The value function $V_R(s_t, t)$ in (14) represents the evolution of the target level set function backwards in time. By finding the solution to $V_R$ in the above PDE, the level set function is determined at any time instant $t \in [-T, 0]$, thereby determining the BRT. From the result of Theorem 2 in Mitchell et al. (2005), it is proved that the solution of $V_R$ in (14) at any time $t$ gives the zero sublevel set for the BRT. Thus,

$$\mathbb{B}_R(-T) = \{s : V_R(s_t, t) \leq 0, \ t \in [-T, 0]\}. \tag{15}$$

The solution to $V_R$ can be computed numerically by using existing solvers for the level set method. A brief note on the implementation of the algorithm is included in subsection A.3 in the Appendix.

There are a few things to note about the formulation in (14). First, Equation (14) assumes that $\mathbb{T}$ is a desired goal state. However, the formulation can be modified if $\mathbb{T}$ is an unsafe set, in which case, the adversarial modeling error tries to minimize the Hamiltonian. Similarly, the input $d$ can represent any other disturbance in the system, either adversarial or cooperative. Second, to compute the FRT, the formulation in (14) is modified from a final value PDE to an initial value PDE.

## 4 EXPERIMENTS

In our experiments, we aim to answer the following questions: **(a)** Can safety verification be done for an NN-based $\pi$ and $\hat{f}_\theta$ using FRT?, and **(b)** Can $\mathbb{S}_{safe}$ be identified using BRT if $\pi$ is deemed unsafe? To answer these two questions, we demonstrate results on the following domains inspired by

real world safety-critical problems, where RL controllers developed using a learnt model can be appealing, as they can adapt to transition dynamics involving friction, air-drag, wind, etc., which might be hard to explicitly model otherwise. It is noted that since the proposed verification framework is developed for control-oriented tasks for physical systems, the state representation of such systems comprises of position, velocity and orientation data. Therefore, the state dimensions of such class of problems are typically not as large as the popular image-based OpenAI or Deepmind domains. Instead, the results are demonstrated on experimental domains which are similar to the ones in prior works on safety verification of NN controllers for physical systems (Xiang et al., 2018b; Xiang & Johnson, 2018; Akintunde et al., 2018; 2019).

**Safe land navigation:** Navigation of a ground robot in an indoor environment is a common application which requires the satisfaction of safety constraints by $\pi$ to avoid collision with obstacles. For this setting, we simulate a ground robot which has continuous states and actions. The initial configuration of the domain is shown in Fig. 2. The set of initial and goal states are represented by circles and the obstacles with rectangles.

**Safe aerial navigation:** This domain simulates a navigation problem in an urban environment for an unmanned aerial vehicle (UAV). Constraints are incorporated while training $\pi$ to ensure that collision is avoided with potential obstacles in its path. States and actions are both continuous and the initial configuration of the domain is shown in Fig 3. The set of initial and goal states are represented using cuboids, and the obstacles with cylinders.

**Analysis:** To address the questions with respect to the above mentioned domains, we first train a NN based $\hat{f}_\theta$ to estimate the dynamics using sampled transitions. This $\hat{f}_\theta$ is then also used to learn a NN based controller $\pi$ which is trained with a cost function designed to mitigate collisions. For brevity, only the representative results for this $\pi$ are discussed here; implementation details and more experimental results are available in Appendix A.2, A.3, A.4 and A.5.

To address the first question, the FRT is computed for both the domains over the augmented learnt dynamics $\hat{f}_\theta^{(r)}$ as shown in Fig. 2 and Fig. 3. Additionally, for land navigation we also compute the FRT over the true system dynamics $f$, which serves as a way to validate the safety verification result of $\pi$ from the proposed framework. It is observed that for both the domains, FRTs deem the given policy $\pi$ as *unsafe*, since the FRTs intersect with one of the obstacles. Even when $\pi$ is learnt using a cost function designed to avoid collisions, the proposed safety verification framework successfully brings out the limitations of $\pi$, which may have resulted due to the use of function approximations, ill-specified hyper-parameters, convergence to a local optimum, etc.

For the second question, the BRT is computed from both the obstacles for the given controller $\pi$, as shown in Fig. 2 and Fig. 3. To estimate the accuracy of the BRT computation, we compare the computed $\mathbb{S}_{unsafe}$ and $\mathbb{S}_{safe}$ sets with the ground truth (G.T.) data generated using random samples of possible trajectories. It is observed that the BRT from obstacle 1 does not intersect with $\mathbb{S}_0$, implying that all trajectories are safe w.r.t. obstacle 1. However, the BRT from obstacle 2 intersects with $\mathbb{S}_0$ and identifies the subset of initial states which are unsafe. The set of unsafe initial states computed by the BRT algorithm may not be exact, as is seen in Fig. 3 where the BRT computation over approximates $\mathbb{S}_{unsafe}$. Such an information can be critical to safely deploy even an unsafe controller just by restricting its starting conditions.

## 5 CONCLUSION

In this paper, we have presented a novel framework using forward and backward reachable tubes for safety verification and determination of the subset of initial states for which a given model-based RL controller always satisfies the state constraints. The main contribution of this work is the formulation of the reachability problem for a neural network modeled system dynamics and the use of level set method to compute an exact reachable tube by solving the Hamilton-Jacobi partial differential equation, for the reinforcement learning framework, thereby minimizing approximation errors that other existing reachability methods suffer. Additionally, the proposed framework can identify the set of safe initial sets for a given policy, thereby determining the initial conditions for which even a sub-optimal, unsafe policy satisfies the safety constraints.

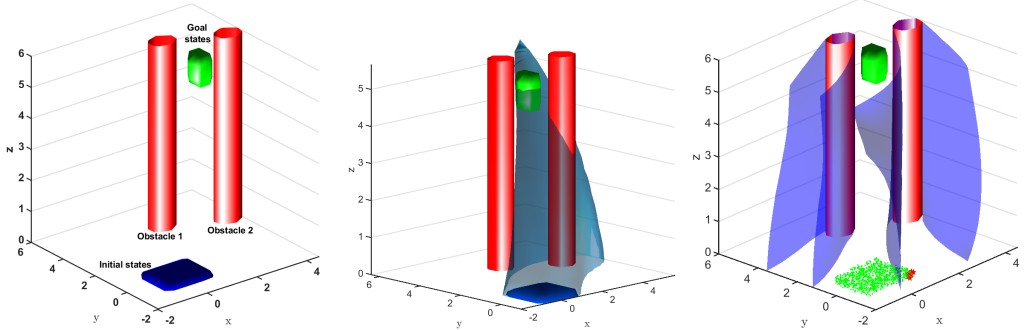

Figure 3: (Left) The safe aerial navigation domain. (Middle) The FRT is computed for the augmented learnt dynamics. For clarity in 3D, FRT with true dynamics is not plotted; instead the approximation quality can be better visualized in BRT. (Right) Comparison of $\mathbb{S}_{safe}$ computed by the proposed BRT algorithm with the ground truth data of the safe initial states, marked in green (and unsafe states marked in red), it is observed that the BRT over approximates $\mathbb{S}_{unsafe}$.

While the results from the proposed framework are promising, there is still room for improvement. One of the drawbacks of using the level set method is the fact that it scales poorly with the increasing dimension of the state space. Recent progress in addressing the scalability issue includes decomposition of system dynamics into subsystems which can later be coupled via common states or controls (Bansal et al., 2017; Margellos & Lygeros, 2011; Chen et al., 2018). Additionally, the application of reachability analysis in developing safe learning based controllers is also a promising direction (Akametalu et al., 2014; Fisac et al., 2018).

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
