# OpenReview forum: "Safety Verification of Model Based Reinforcement Learning Controllers"
_ICLR.cc/2021/Conference — Reject_

### Official Review · AnonReviewer2 · 2020-10-25
**Review of Safety Verification of Model-based RL Controllers**

**Rating:** 3
**Confidence:** 4

**Review:**

Summary: The paper studies the problem of safety verification of model-based RL controllers in continuous action&state spaces. The paper proposes a verification method based on reachable set analysis. The proposed method either verifies the controller to be safe or identifies the subset of initial states that are safe.

Strengths:

i) The motivation, organization and the overall writing of the paper are clear.

Weakness:

i) The problem of verifying properties of NN based controllers is studied in the literature [1,2,3,4,5,6,7,8,9,10,11,12,…], which is not discussed in detail in the paper (i.e., there is a paragraph present in the supplementary material). Moreover, there are no experimental comparisons made to any of these previous works.

ii) The tested experimental domains are not the best representatives of the control problems motivated in the paper. They are extremely small in size and do not really require NN approximation. As such, the resulting NN architectures are very small (as seen in the supplementary materials).

References:

[1] Verisig: verifying safety properties of hybrid systems with neural network controllers, Ivanov et al. HSCC-19.

[2] Learning and Verification of Feedback Control Systems using Feedforward Neural Networks, Dutta et al., 2018.

[3] Reachability Analysis and Safety Verification for Neural Network Control Systems, Xiang and Johnson, arvix 2018.

[4] Efficient Verification of Control Systems with Neural Network Controllers, Yang et al., ICVISP-19.

[5] Verifying Deep-RL-Driven Systems, Kazak et al. NetAI-19.

[6] Guarded Deep Learning using Scenario-based Modeling, Katz MODELSWARD 2020.

[7] Reachability Analysis for Neural Agent-Environment Systems, Akintunde et al, KR-18.

[8] Verification of RNN-Based Neural Agent-Environment Systems, Akintunde et al, AAAI-19.

[9] Reachability Analysis for Neural Feedback Systems using Regressive Polynomial Rule Inference, Dutta et al., HSCC-19.

[10] ReachNN: Reachability Analysis of Neural-Network Controlled Systems, Huang et al., TECS-19.

[11] Reachable Set Estimation and Safety Verification for Piecewise Linear Systems with Neural Network Controllers, Xiang et al., ACC-18.

[12] A Reachability Method for Verifying Dynamical Systems with Deep Neural Network Controllers, Julian et al., arvix, 2019.

---

> ### Author Response · Authors · 2020-11-17
> **Rebuttal reply 1**
>
> We thank you for your insightful suggestions. In the following, we address the questions raised (Due to the character limit, we have divided our response in two separate comments).
>
> Q1: "Update literature to compare with relevant work"
> A1: Thank you for providing these additional references. Indeed, as you mentioned, there has been substantial work done in verifying properties of NN using forward reachable set analysis (different from our backward reachable set analysis) and we had kept the discussion short due to the page limit. But we now see that readers can benefit from some additional discussion and we have updated the “Extended related work” section in Appendix Section A.1 to discuss the mentioned papers [1-12] in more detail.
>
> To summarize, the proposed algorithm is distinct from the current literature because it is applicable to a general class of activation functions (see Assumption 1), whereas the verification algorithms in [2], [4], [7], [8], [11] are designed for ReLU activation functions and in [1] is designed for sigmoidal DNNs.  The algorithms developed in [3], [9], [10] and [12] are applicable for a general class of activation functions, however, they only compute the forward reachable tube to classify a neural network as safe or unsafe. In comparison, one of the main contributions of this work is the computation of the backward reachable tube, leading to not only the classification of a neural network controller as safe or unsafe but also the determination of safe initial states – which hasn’t been explored in other works, to the best of our knowledge.
>
>
> References:
> [1] Ivanov, Radoslav, James Weimer, Rajeev Alur, George J. Pappas, and Insup Lee. "Verisig: verifying safety properties of hybrid systems with neural network controllers." In Proceedings of the 22nd ACM International Conference on Hybrid Systems: Computation and Control, pp. 169-178. 2019.
>
> [2] Dutta, Souradeep, Susmit Jha, Sriram Sankaranarayanan, and Ashish Tiwari. "Learning and verification of feedback control systems using feedforward neural networks." IFAC-PapersOnLine 51, no. 16 (2018): 151-156.
>
> [3] Xiang, Weiming, and Taylor T. Johnson. "Reachability analysis and safety verification for neural network control systems." arXiv preprint arXiv:1805.09944 (2018).
>
> [4] Yang, Guoqing, Guangyi Qian, Pan Lv, and Hong Li. "Efficient Verification of Control Systems with Neural Network Controllers." In Proceedings of the 3rd International Conference on Vision, Image and Signal Processing, pp. 1-7. 2019.
>
> [5] Kazak, Yafim, Clark Barrett, Guy Katz, and Michael Schapira. "Verifying deep-RL-driven systems." In Proceedings of the 2019 Workshop on Network Meets AI & ML, pp. 83-89. 2019.
>
> [6] Katz, Guy. "Guarded Deep Learning using Scenario-Based Modeling." arXiv preprint arXiv:2006.03863 (2020).
>
> [7] Akintunde, Michael, Alessio Lomuscio, Lalit Maganti, and Edoardo Pirovano. "Reachability Analysis for Neural Agent-Environment Systems." In KR, pp. 184-193. 2018.
>
> [8] Akintunde, Michael E., Andreea Kevorchian, Alessio Lomuscio, and Edoardo Pirovano. "Verification of rnn-based neural agent-environment systems." In Proceedings of the AAAI Conference on Artificial Intelligence, vol. 33, pp. 6006-6013. 2019.
>
> [9] Dutta, Souradeep, Xin Chen, and Sriram Sankaranarayanan. "Reachability analysis for neural feedback systems using regressive polynomial rule inference." In Proceedings of the 22nd ACM International Conference on Hybrid Systems: Computation and Control, pp. 157-168. 2019.
>
> [10] Huang, Chao, Jiameng Fan, Wenchao Li, Xin Chen, and Qi Zhu. "ReachNN: Reachability analysis of neural-network controlled systems." ACM Transactions on Embedded Computing Systems (TECS) 18, no. 5s (2019): 1-22.
>
> [11] Xiang, Weiming, Hoang-Dung Tran, Joel A. Rosenfeld, and Taylor T. Johnson. "Reachable set estimation and safety verification for piecewise linear systems with neural network controllers." In 2018 Annual American Control Conference (ACC), pp. 1574-1579. IEEE, 2018.
>
> [12] Julian, Kyle D., and Mykel J. Kochenderfer. "A reachability method for verifying dynamical systems with deep neural network controllers." arXiv preprint arXiv:1903.00520 (2019).

---

> ### Author Response · Authors · 2020-11-17
> **Rebuttal reply 2**
>
> Q2: "Size of tested experimental domains"
> A2:
> We now see that some more discussion around the motivation of the experimental setup could have been beneficial for the readers. We’d like to note that application of NN’s to solving control problems has shown promising results in environments where uncertainties cannot be explicitly modeled or incorporated a priori in the system dynamics (e.g., friction forces for land robots, wind disturbances for UAVs, water turbulence during underwater exploration, etc.) [17, 22]. Therefore, the simulation problems presented in the paper are motivated by widely considered control problems for robotic systems using NN-based controllers, including robotic arms [13, 14, 15] and UAVs [16, 17, 18, 19], amongst other navigation systems [20, 21]. These works typically represent the system’s state using its position, velocity and orientation data, which don’t exceed a high number – for instance, prior works have used under 12 state dimensions to model the state of UAV’s [17, 18, 19]. Thus, the size of NN architectures in these applications are much smaller than say image-based inputs, and can be tackled by the proposed verification algorithm. Additionally, we will remind the readers that some related works on safety verification have also evaluated their forward reachable set analysis on physical systems and the size of NN architectures evaluated in this paper are similar to these prior works [7, 8, 10, 11]. We should have, and will, duly include this clarification in Section 4 of the main paper
>
> Finally, despite the small NN architecture, the proposed algorithm extends the state-of-the-art verification methods by computing the backward reachable tube to determine the set of unsafe initial states in addition to classifying a neural network based controller as safe or unsafe – a problem which has not been addressed in previous works, to the best of our knowledge.
>
> References:
> [13] Kumar, Vikash, Emanuel Todorov, and Sergey Levine. "Optimal control with learned local models: Application to dexterous manipulation." In 2016 IEEE International Conference on Robotics and Automation (ICRA), pp. 378-383. IEEE, 2016.
>
> [14] Gu, Shixiang, Ethan Holly, Timothy Lillicrap, and Sergey Levine. "Deep reinforcement learning for robotic manipulation with asynchronous off-policy updates." In 2017 IEEE international conference on robotics and automation (ICRA), pp. 3389-3396. IEEE, 2017.
>
> [15] Gupta, Abhishek, Clemens Eppner, Sergey Levine, and Pieter Abbeel. "Learning dexterous manipulation for a soft robotic hand from human demonstrations." In 2016 IEEE/RSJ International Conference on Intelligent Robots and Systems (IROS), pp. 3786-3793. IEEE, 2016.
>
> [16] Hwangbo, Jemin, Inkyu Sa, Roland Siegwart, and Marco Hutter. "Control of a quadrotor with reinforcement learning." IEEE Robotics and Automation Letters 2, no. 4 (2017): 2096-2103.
>
> [17] Xiang, Tian, Fan Jiang, Qi Hao, and Wang Cong. "Adaptive flight control for quadrotor UAVs with dynamic inversion and neural networks." In 2016 IEEE International Conference on Multisensor Fusion and Integration for Intelligent Systems (MFI), pp. 174-179. IEEE, 2016.
>
> [18] Kim, Boo Min, Kwang Chan Choi, and Byoung Soo Kim. "Trajectory tracking controller design using neural networks for tiltrotor UAV." In AIAA Guidance, Navigation and Control Conference and Exhibit, p. 6460. 2007.
>
> [19] Clarke, Shanelle G., and Inseok Hwang. "Deep Reinforcement Learning Control for Aerobatic Maneuvering of Agile Fixed-Wing Aircraft." In AIAA Scitech 2020 Forum, p. 0136. 2020.
>
> [20] Haarnoja, Tuomas, Aurick Zhou, Kristian Hartikainen, George Tucker, Sehoon Ha, Jie Tan, Vikash Kumar et al. "Soft actor-critic algorithms and applications." arXiv preprint arXiv:1812.05905 (2018).
>
> [21] ALTUNTAŞ, NİHAL, Erkan Imal, Nahit Emanet, and Ceyda Nur Öztürk. "Reinforcement learning-based mobile robot navigation." Turkish Journal of Electrical Engineering & Computer Sciences 24, no. 3 (2016): 1747-1767.
>
> [22] Johannink, Tobias, Shikhar Bahl, Ashvin Nair, Jianlan Luo, Avinash Kumar, Matthias Loskyll, Juan Aparicio Ojea, Eugen Solowjow, and Sergey Levine. "Residual reinforcement learning for robot control." In 2019 International Conference on Robotics and Automation (ICRA), pp. 6023-6029. IEEE, 2019.

---

> ### Author Response · Authors · 2020-11-24
> **Updated manuscript**
>
> We thank you for your comments.
>
> We have updated Section A.1 - Extended Literature Review, to include the references mentioned in Q1. Additionally, towards the end of first paragraph in Section 4 - Experiments, we have mentioned that the experimental domains considered in this work are comparable to similar prior works and have duly cited those papers.

---

### Official Review · AnonReviewer4 · 2020-10-28
**Verification of model-based RL using reachable set analysis**

**Rating:** 7
**Confidence:** 2

**Review:**

# Summary

The paper presents a method for safety verification of trained model-based RL agents using reachable set analysis.
The method allows to both analyse whether any of the start states might end in an unsafe state, via the forward reachable tube, and to determine which start states do or do not end in an unsafe state, via the backward reachable tube.
Another claimed contribution is the first method to calculate the backward reachable tube on NNs.
An experimental evaluation is performed on to navigation environments.

# Comments to the Authors

Safety verification is a crucial topic for the acceptance and deployment of learned methods in control systems.
This paper contributes a method to verify a model-based RL controller in forwards and backwards direction, given the trained controller, a set of start and goal states and the constraints, i.e. a description of the unsafe states.

The method appears sound and is presented as straightforward, which will allow adoption in future works and potentially pratical application.
It includes the transfer of the backward reachable tube computation to NNs, which is central to the method.
Since it is outside my expertise, I do not verify the theoretical contributions of the paper.

Coming from the experimental evaluation, there are a few questions regarding the applicability of the method:
a) How would it handle dynamic environments, e.g. moving obstacles? As it is, only the deterministic trajectory of the agent appears to be evaluated.
b) How complex is the method computationally? What are the runtimes for the given examples?
c) In Figure 3 (right) it appears as if some green starting states are in an unsafe area. Is this a visual artifact or does the method not guarantee that all unsafe starting states are discovered (which was my impression before)?

Regarding the presentation of the paper, I understand that the authors are restricted by the conference format, but I would encourage to revise the selection of material for the main paper and the appendices.
The related work part of the paper might be benefit from some of the extended work in the appendix and the experiments could also be described more in detail.
At the same time, maybe the size of the flowchart could be improved or non-crucial parts of Sec. 3.2 extracted to the appendix.
Also, I think it is okay to put the paper + appendix as the main PDF on openreview, which makes the appendices more accessible for the readers on this platform.

---

> ### Author Response · Authors · 2020-11-17
> **Rebuttal reply 1**
>
> Thank you for your valuable suggestions. We have addressed the questions below.
>
> Q1: "Dynamic environment"
>
> A1: Currently, the proposed algorithm is developed for an environment. Since this is our first attempt at solving the backward reachable tube problem for NN-based model dynamics, we have assumed a stationary environment in this paper. However, we’d like to note that this assumption is commonly used and similar to the problem settings considered in related works on safety verification of NN-based controllers [1, 2, 3, 4]. We are currently extending the proposed algorithm to dynamic environments.
>
> Q2: "Computational complexity"
>
> A2: The proposed algorithm will not scale well for large scale problems such as image-based input systems. However, the proposed algorithm is developed for physical systems (robotic systems such as ground vehicles, UAVs, robotic arms etc.), where the system’s state representation comprises of position, velocity and orientation data. The order of state dimensions of such systems is usually around ten. For example, a UAV’s complete state information can be provided using 12 state dimensions [5, 6], but depending on specific problems, prior works have used a lesser number of state dimensions as well [7], and a comparable number of state dimensions is used for developing NN controllers for robotic arms [8, 9, 10]. Thus, the proposed verification algorithm can tackle the problems of interest. We would also like to note that some related works on safety verification have also evaluated their forward reachable set analysis on physical systems and the size of NN architectures evaluated in this paper are similar to these prior works [1, 2, 3, 4].
> Additionally, there are independent works which are looking into addressing the scalability issue for nonlinear system dynamics with higher dimensions, by decoupling the system states and decomposing the system into smaller components [11, 12]. A potential direction to improve the scalability of the proposed algorithm is to leverage this decoupling technique for NNs and divide the overall reachability problem into smaller problems. Perhaps we can add this discussion in a “Future Work” section in the main paper.
>
> References:
> [1] Xiang, Weiming, Hoang-Dung Tran, Joel A. Rosenfeld, and Taylor T. Johnson. "Reachable set estimation and safety verification for piecewise linear systems with neural network controllers." In 2018 Annual American Control Conference (ACC), pp. 1574-1579. IEEE, 2018.
>
> [2] Akintunde, Michael, Alessio Lomuscio, Lalit Maganti, and Edoardo Pirovano. "Reachability Analysis for Neural Agent-Environment Systems." In KR, pp. 184-193. 2018.
>
> [3] Akintunde, Michael E., Andreea Kevorchian, Alessio Lomuscio, and Edoardo Pirovano. "Verification of rnn-based neural agent-environment systems." In Proceedings of the AAAI Conference on Artificial Intelligence, vol. 33, pp. 6006-6013. 2019.
>
> [4] Xiang, Weiming, and Taylor T. Johnson. "Reachability analysis and safety verification for neural network control systems." arXiv preprint arXiv:1805.09944 (2018).
>
> [5] Xiang, Tian, Fan Jiang, Qi Hao, and Wang Cong. "Adaptive flight control for quadrotor UAVs with dynamic inversion and neural networks." In 2016 IEEE International Conference on Multisensor Fusion and Integration for Intelligent Systems (MFI), pp. 174-179. IEEE, 2016.
>
> [6] Clarke, Shanelle G., and Inseok Hwang. "Deep Reinforcement Learning Control for Aerobatic Maneuvering of Agile Fixed-Wing Aircraft." In AIAA Scitech 2020 Forum, p. 0136. 2020.
>
> [7] Kim, Boo Min, Kwang Chan Choi, and Byoung Soo Kim. "Trajectory tracking controller design using neural networks for tiltrotor UAV." In AIAA Guidance, Navigation and Control Conference and Exhibit, p. 6460. 2007.
>
> [8] Kumar, Vikash, Emanuel Todorov, and Sergey Levine. "Optimal control with learned local models: Application to dexterous manipulation." In 2016 IEEE International Conference on Robotics and Automation (ICRA), pp. 378-383. IEEE, 2016.
>
> [9] Gu, Shixiang, Ethan Holly, Timothy Lillicrap, and Sergey Levine. "Deep reinforcement learning for robotic manipulation with asynchronous off-policy updates." In 2017 IEEE international conference on robotics and automation (ICRA), pp. 3389-3396. IEEE, 2017.
>
> [10] Gupta, Abhishek, Clemens Eppner, Sergey Levine, and Pieter Abbeel. "Learning dexterous manipulation for a soft robotic hand from human demonstrations." In 2016 IEEE/RSJ International Conference on Intelligent Robots and Systems (IROS), pp. 3786-3793. IEEE, 2016.
>
> [11] Chen, Mo, Jennifer C. Shih, and Claire J. Tomlin. "Multi-vehicle collision avoidance via Hamilton-Jacobi reachability and mixed integer programming." In 2016 IEEE 55th Conference on Decision and Control (CDC), pp. 1695-1700. IEEE, 2016.
>
> [12] Chen, Mo, Sylvia Herbert, and Claire J. Tomlin. "Exact and efficient Hamilton-Jacobi guaranteed safety analysis via system decomposition." In 2017 IEEE International Conference on Robotics and Automation (ICRA), pp. 87-92. IEEE, 2017.

---

> > ### Comment · AnonReviewer4 · 2020-11-18
> > **Response**
> >
> > Thank you for the explanations. I understand the limitations on the computational complexity and it makes sense given the envisioned use case.
> > Regarding the dynamic environments, I still think this is a limitation, but I also accept that it seems to be a common limitation in the current literature.

---

> ### Author Response · Authors · 2020-11-17
> **Rebuttal reply 2**
>
> Q3: "Clarification for Figure 3"
>
> A3: No, it is not a visual artifact. The proposed method indeed identifies all the unsafe starting states, which are marked in red in Figure 3. While the reachable set computed by the proposed algorithm is exact for the learnt model, f(r), due to a conservative bound on the modeling error ‘d’, the unsafe initial set is over-approximated and some safe initial states, marked in green, are falsely classified as unsafe. However, for safety critical systems, such over-approximation of the unsafe states is preferable to under-approximation since this property ensures safety even for a system with modeling error. We will add this point in the caption for Figure 3.

---

> > ### Comment · AnonReviewer4 · 2020-11-18
> > **Clarification**
> >
> > Thanks for the clarification. I agree that you would want to over-approximate. Having a clearer caption would be helpful here.
> >
> > My main surprise was that in Fig. 3 (right) some of the green safe states are to the right of the BRT, which I interpreted such that they would be in the "unsafe territory" since the BRT divides the space.

---

> > > ### Author Response · Authors · 2020-11-24
> > > **Updated manuscript**
> > >
> > > We thank you for your comments.
> > > We have modified the manuscript and added a second paragraph to Section 5 - Conclusion, where we mention the scalability issue of the proposed reachability algorithm and cite some ongoing works to address this issue. We have also modified the caption of Figure 3 (Right) and added a comment in Section 4 - Experiments (Analysis paragraph), that the set of unsafe initial states computed by the algorithm may not be exact and that it over approximates the solution for the safe aerial navigation problem.

---

### Official Review · AnonReviewer1 · 2020-10-29
**Interesting paper, questions about correctness, efficiency, related work, and experiments**

**Rating:** 7
**Confidence:** 4

**Review:**

update after rebuttal: the answers were convincing, and the paper improved.

Summary and Contribution

This work presents a novel approach for the safety verification of model based RL controllers. It uses a so-called reachability tube analysis to check whether unsafe states may be reached from the initial states of a (learned) model. In case unsafe states may be reached, a backward analysis determines the initial states that are safe (if they exist). Existing methods to measure potential modeling errors are taken into account. The efficacy of the method is demonstrated on two small examples.



Reasons for Score

With more details on the correctness, better literature study, and more experiments, the paper would be a clear accept. As of now, I see it as marginally above the acceptance threshold.

Strengths

- A novel method for verifying the safety of NN-based controllers
- A framework to incorporate inherent modeling errors

Weaknesses

- A better comparison to related work is in order
- The experiments are not very extensive, and it is not clear how to reproduce them.
- The correctness and completeness of the method, as well as the concrete assumptions on the NN architecture is unclear to me.

Questions for Authors

- Please compare your work to the literature listed below. What is novel, how does it relate?
- What assumptions are being made on the neural network architecture? How can you obtain an exact and efficient method (soundness, completeness, global optima) for an potentially undecidable problem?
- How would your method perform on state-of-the-art benchmark sets like from OpenAI or Deepmind?
- If you assume model-based RL, and the model is already there, why can't you construct a simpler model like a Markov decision process and reason directly about safety using sound and efficient methods like value iteration or linear programming? I understand you have a continuous state space (as given by the neural network), but do the experiments/examples require such a complicated model?

Detailed Comments

- related work

There is large body of related work on safety of (model-based) RL which has not been cited, for instance:


[1] Prashanth L, A., and Michael Fu. "Risk-sensitive reinforcement learning: A constrained optimization viewpoint." arXiv preprint arXiv:1810.09126 (2018)

[2] Zheng, Liyuan, and Lillian J. Ratliff. "Constrained upper confidence reinforcement learning." arXiv preprint arXiv:2001.09377 (2020).

[3] Eriksson, Hannes, and Christos Dimitrakakis. "Epistemic Risk-Sensitive Reinforcement Learning." arXiv preprint arXiv:1906.06273 (2019).

[4] Sebastian Junges, Nils Jansen, Christian Dehnert, Ufuk Topcu, Joost-Pieter Katoen:
Safety-Constrained Reinforcement Learning for MDPs. TACAS 2016: 130-146

[5] Nils Jansen, Bettina Könighofer, Sebastian Junges, Alex Serban, Roderick Bloem:
Safe Reinforcement Learning Using Probabilistic Shields. CONCUR 2020: 3:1-3:16

[6] Guy Avni, Roderick Bloem, Krishnendu Chatterjee, Thomas A. Henzinger, Bettina Könighofer, Stefan Pranger:
Run-Time Optimization for Learned Controllers Through Quantitative Games. CAV (1) 2019: 630-649

[7] Mohammadhosein Hasanbeig, Alessandro Abate, Daniel Kroening:
Cautious Reinforcement Learning with Logical Constraints. AAMAS 2020: 483-491


[8] Mohammadhosein Hasanbeig, Alessandro Abate, Daniel Kroening:
Logically-Correct Reinforcement Learning. CoRR abs/1801.08099 (2018)


- correctness and efficiency

I don't really understand how you can incorporate nonconvex boundaries of the reachable tube. In any case, it should be an overapproximation. How is the efficiency of the method? How will it perform in practice?

- experiments

It would be very nice to see how the method performs on well-known environments like the OpenAI safety gymn or the Deepmind safety gridworlds.

---

> ### Author Response · Authors · 2020-11-17
> **Rebuttal reply 1**
>
> We thank you for your insightful suggestions. In the following, we address the questions raised (Due to the character limit, we have divided our response in two separate comments).
>
> Q1: "Comparison to literature listed below"
>
> A1: Thank you for pointing out the relevant work. We will update the literature review in Section 1 of the manuscript to compare the proposed algorithm with these works.
>
> We believe that the relevant works cited below are trying to incorporate formal safety in RL algorithms from a different perspective than the work presented in this paper. These works [1,2,3,4,5,6,7,8] incorporate safety constraints in the learning phase to train a controller (policy) to satisfy certain desired specifications or constraints. Whereas our proposed verification algorithm makes no assumptions on how the model-based RL controller is trained. Instead, it works independently of the training phase to identify the safe initial conditions for any given controller (policy).
>
> We will also highlight that one major advantage of using a standalone verification framework is that we can deploy potentially unsafe policies on real systems, without further training, by restricting their initial conditions to only the safe states. Therefore, different from the above references, the proposed algorithm doesn’t make strict assumptions on the complete or accurate knowledge of the system dynamics [2, 7], which can be difficult to obtain.
>
> Further, to incorporate safety during learning, some works approximate the original problem to represent safety constraints in a tractable form [1, 6], which reduces the performance of the final trained controllers [1, 3, 4, 5]. In comparison, our proposed verification algorithm can work on a general class of state constraints which can be represented using level set functions (see Assumption 2) and doesn’t approximate the state constraints.
>
> Q2: "Assumptions on NN architecture & nature of solution"
>
> A2: As you correctly mentioned, obtaining an exact safe region might be intractable under no assumptions. Hence, we do require the neural network to be bounded and Lipschitz smooth, as  discussed under Assumption 1. Further, we will highlight in Section 4 to remind the readers that our identified safe regions need not be exact as we compute conservative model error bounds. This is useful as it allows controlling the approximation error to over-approximate unsafe region such that precision of the identified unsafe states is high.
>
>
> References:
>
> [1] Prashanth L, A., and Michael Fu. "Risk-sensitive reinforcement learning: A constrained optimization viewpoint." arXiv (2018): arXiv-1810.
>
> [2] Zheng, Liyuan, and Lillian J. Ratliff. "Constrained upper confidence reinforcement learning." arXiv preprint arXiv:2001.09377 (2020).
>
> [3] Eriksson, Hannes, and Christos Dimitrakakis. "Epistemic Risk-Sensitive Reinforcement Learning." arXiv preprint arXiv:1906.06273 (2019).
>
> [4] Junges, Sebastian, Nils Jansen, Christian Dehnert, Ufuk Topcu, and Joost-Pieter Katoen. "Safety-constrained reinforcement learning for MDPs." In International Conference on Tools and Algorithms for the Construction and Analysis of Systems, pp. 130-146. Springer, Berlin, Heidelberg, 2016.
>
> [5] Könighofer, Bettina, Roderick Bloem, Sebastian Junges, Nils Jansen, and Alex Serban. "Safe Reinforcement Learning Using Probabilistic Shields." In International Conference on Concurrency Theory: 31st CONCUR 2020: Vienna, Austria (Virtual Conference). Schloss Dagstuhl-Leibniz-Zentrum fur Informatik GmbH, Dagstuhl Publishing, 2020.
>
> [6] Avni, Guy, Roderick Bloem, Krishnendu Chatterjee, Thomas A. Henzinger, Bettina Könighofer, and Stefan Pranger. "Run-time optimization for learned controllers through quantitative games." In International Conference on Computer Aided Verification, pp. 630-649. Springer, Cham, 2019.
>
> [7] Hasanbeig, Mohammadhosein, Alessandro Abate, and Daniel Kroening. "Cautious reinforcement learning with logical constraints." arXiv preprint arXiv:2002.12156 (2020).
>
> [8] Hasanbeig, Mohammadhosein, Alessandro Abate, and Daniel Kroening. “Logically-Correct Reinforcement Learning”. CoRR abs/1801.08099 (2018).

---

> ### Author Response · Authors · 2020-11-17
> **Rebuttal reply 2**
>
> Q3: "Performance on OpenAI or Deepmind benchmark sets"
>
> A3: We’d like to note that our method is developed for control-oriented tasks for physical systems, like navigation of a robot or a vehicle on land, in the air, or on water, and thus the benchmark sets from OpenAI or Deepmind may not be the best representatives for  the problem considered in this work. For example, the experimental domains considered in this paper are similar to the ones in prior works on safety verification of NN controllers for physical systems [9, 10, 11, 12], where the forward reachable set analysis was conducted on NN architectures of comparable sizes. Additionally, since the state representation of physical systems (like UAVs, ground vehicles, robotic arms etc.) comprise of position, velocity and orientation data, their state dimensions typically are not as large as the popular image-based OpenAI or Deepmind domains (for instance, prior works have used under 12 state dimensions to model the state of UAV’s [13, 14, 15]). Therefore, the proposed verification algorithm is suitable to address the problems of interest. We will add this discussion in Section 4 of the main paper.
>
> Q4: "Constructing a second MDP and solving the problem using value iteration"
>
> A4: Thank you for asking this. We now see that some more discussion at the start of Section 3 could have been beneficial for the readers. Doing value iteration directly on the MDP induced from the NN model could indeed provide a reasonable policy, but the resulting policies need not be safe. If the MDP was discrete, or close to deterministic, or had linear dynamics, one could potentially then use this policy to roll out random trajectories in the learned MDP to reason about safety violation. However, this approach would be impractical if the states and actions are both continuous (which is the case for physical systems such as robots and vehicles) and/or if the system dynamics of physical systems are nonlinear and have several sources of uncertainties (e.g., friction forces for land robots, wind disturbances for UAVs, water turbulence during underwater exploration, etc.).
> Therefore, the proposed algorithm uses the approach of reachability analysis to efficiently propagate a set of trajectories in the continuous domain while following nonlinear system dynamics (NN-based model), and computing the safe initial conditions. The HJB framework used in the proposed algorithm can also handle uncertainties in the system dynamics and can thus be tractably applied  to the class of physical problems considered in this work.
>
>
> References:
>
> [9] Xiang, Weiming, Hoang-Dung Tran, Joel A. Rosenfeld, and Taylor T. Johnson. "Reachable set estimation and safety verification for piecewise linear systems with neural network controllers." In 2018 Annual American Control Conference (ACC), pp. 1574-1579. IEEE, 2018.
>
> [10] Akintunde, Michael, Alessio Lomuscio, Lalit Maganti, and Edoardo Pirovano. "Reachability Analysis for Neural Agent-Environment Systems." In KR, pp. 184-193. 2018.
>
> [11] Akintunde, Michael E., Andreea Kevorchian, Alessio Lomuscio, and Edoardo Pirovano. "Verification of rnn-based neural agent-environment systems." In Proceedings of the AAAI Conference on Artificial Intelligence, vol. 33, pp. 6006-6013. 2019.
>
> [12] Xiang, Weiming, and Taylor T. Johnson. "Reachability analysis and safety verification for neural network control systems." arXiv preprint arXiv:1805.09944 (2018).
>
> [13] Xiang, Tian, Fan Jiang, Qi Hao, and Wang Cong. "Adaptive flight control for quadrotor UAVs with dynamic inversion and neural networks." In 2016 IEEE International Conference on Multisensor Fusion and Integration for Intelligent Systems (MFI), pp. 174-179. IEEE, 2016.
>
> [14] Kim, Boo Min, Kwang Chan Choi, and Byoung Soo Kim. "Trajectory tracking controller design using neural networks for tiltrotor UAV." In AIAA Guidance, Navigation and Control Conference and Exhibit, p. 6460. 2007.
>
> [15] Clarke, Shanelle G., and Inseok Hwang. "Deep Reinforcement Learning Control for Aerobatic Maneuvering of Agile Fixed-Wing Aircraft." In AIAA Scitech 2020 Forum, p. 0136. 2020

---

> > ### Author Response · Authors · 2020-11-24
> > **Updated manuscript**
> >
> > We thank you for your comments. We have modified the manuscript and updated the literature review in Section 1 - Introduction to include the references mentioned in Q1. Additionally, we have specified in Section 4 - Experiments (Analysis paragraph) that the solution computed by the BRT algorithm may not be exact, as discussed in Q2. We have also explained that the problem domains in OpenAI or Deepmind are not the best representatives of the problems considered in this work. This is included towards the end of the first paragraph in Section 4 - Experiments.

---

### Official Review · AnonReviewer3 · 2020-10-29
**ICLR 2021**

**Rating:** 5
**Confidence:** 3

**Review:**

## Summary

The authors propose a safety verification algorithm based on reachability analysis and formal methods that is applicable to dynamics models and policies that are parameterized as neural networks.

I find the problem(s) clear and well-motivated in general. There are many prior works in this area from outside the machine learning community, and I'm not totally convinced that this paper provides a substantial contribution over those works.

## Strengths

 - The problem setup is clear and well-motivated
 - The methodology seems solid

## Weaknesses

 - The work is framed in the context of model-based RL, but since the model (including the estimate of the modeling error bound $d$) is assumed known, it would seem like this should be framed more squarely within the optimal control literature. Within this literature, there are many methods related to Safe MPC (e.g. [1], [2])  that it would seem are attempting to achieve the same objective as what is stated here.

 - The central issue with all of the host of works on formal methods and reachability analysis for dynamical systems is that they don't scale to complex environments or dynamics. As a result, assumptions need to made such as monotonicity (work of Del Vecchio e.g. [3] [4]). The experiments that you show are very simplistic also (low dimensional and very simple/convex geometry). Is there something about this approach which allows us to make progress in terms of scaling these methods to real world problems.

 - You argue that "Estimating modeling errors is an active area of research and is required for several existing works on safe RL (Akametalu et al., 2014; Gillula & Tomlin, 2012), and is complementary to our goal. Since the primary contribution of this work is the development of a reachable tube formulation for model-based controllers that use NNs, we rely on existing techniques (Moldovan et al., 2015) to estimate a conservative modeling error bound." but I find this assumption that you are given the modeling error bound diminishes the contribution of the work since everything downstream in your algorithm will be impacted by this value, which is hard to get in practice.

## Minor Comments/Questions

 - Similar to comment above, what is a "model-based policy"? If you mean that it is derived from model-based RL method, why is this important?

- It would seem that taking the worst case modeling error, while able to provide guarantees (subject to the accuracy of the modeling error), also results in conservatism. A comment on this in the manuscript would be nice.

- Part of the novelty that is claimed here is that the dynamics and policy are parametrized by neural networks, but what is the limitation of previous works on reachability analysis that would not permit a NN model to be used?

[1] Learning-based Model Predictive Control for Safe Exploration
T Koller, F Berkenkamp, M Turchetta, A Krause
Decision and Control (CDC), 2018 IEEE 57th Conference on

[2] Safe nonlinear trajectory generation for parallel autonomy with a dynamic vehicle model
W Schwarting, J Alonso-Mora, L Paull, S Karaman, D Rus
IEEE Transactions on Intelligent Transportation Systems 19 (9), 2994-3008

[3] M. R. Hafner and D. Del Vecchio, “Computational tools for the safety control of a class of piecewise continuous systems with imperfect information on a partial order,” SIAM J. Control Optim., vol. 49, pp. 2463–2493, 2011.

[4] R. Ghaemi and D. D. Vecchio, “Control for safety specifications of systems
with imperfect information on a partial order,

---

> ### Author Response · Authors · 2020-11-17
> **Rebuttal reply 1**
>
> We thank you for your insightful suggestions. In the following, we address the questions raised (Due to the character limit, we have divided our response into separate comments).
>
> Q1 (Weaknesses part 1): "Assumption on modeling error and comparison to Safe MPC based works"
>
> A1:
> We’d like to note that the bounds on the modeling error ‘d’ for our experiments are not assumed to be known, but are computed using a confidence interval computed over the validation dataset of the learnt model. Similarly, we don’t assume any a priori knowledge of the system dynamics to construct the model. The proposed verification algorithm uses the learnt NN-based model which is constructed while developing the model-based RL controller. We will highlight these points in Section 4 and Section 3, respectively.
>
> We will also clarify that in comparison to safe MPC based works, as stated above, we don’t assume any prior, physics-based knowledge of the system dynamics, as is the case in [1]. Also, [1] assumes that the system is subjected to polytopic constraints, whereas our proposed verification algorithm relaxes this assumption and can work on constraint sets which can be represented using a level set function (see Assumption 2). A different class of safe MPC based works assume human inputs in the control loop [2]. This is also an interesting direction, but we believe that this class of problems is significantly different from the ones considered in this work as we don’t assume any form of human input.
>
> References:
>
> [1] Koller, Torsten, Felix Berkenkamp, Matteo Turchetta, and Andreas Krause. "Learning-based model predictive control for safe exploration." In 2018 IEEE Conference on Decision and Control (CDC), pp. 6059-6066. IEEE, 2018.
>
> [2] Schwarting, Wilko, Javier Alonso-Mora, Liam Paull, Sertac Karaman, and Daniela Rus. "Safe nonlinear trajectory generation for parallel autonomy with a dynamic vehicle model." IEEE Transactions on Intelligent Transportation Systems 19, no. 9 (2017): 2994-3008.

---

> ### Author Response · Authors · 2020-11-17
> **Rebuttal reply 2**
>
> Q2 (Weaknesses part 2): "Scalability of reachability algorithm"
>
> A2:
> Your observation is correct, like most methods on reachability analysis, the proposed algorithm doesn’t scale well with increasing state dimensions. However, the problem domain considered in this work is the physical system (e.g., robotic systems such as ground vehicles, UAVs, robotic arms etc.), where the system’s state representation typically comprises of position, velocity and orientation data. Therefore, for such systems, the order of state dimensions is usually around ten. For example, a UAV’s complete state information can be provided using 12 state dimensions [3, 4], but depending on specific problems, prior works have used even lesser number of state dimensions as well [5], and a comparable number of state dimensions is used for developing NN controllers for robotic arms [6, 7, 8]. Thus, the proposed verification algorithm can tackle these problems of interest.
>
> Additionally, there are independent works which are looking into addressing the scalability issue for nonlinear system dynamics with higher dimensions, by decoupling the system states and decomposing the system into smaller components [9, 10]. A potential future direction to improve the scalability of proposed algorithm could be to leverage this decoupling technique for NNs and divide the overall reachability problem into smaller problems. We can add this discussion in a “Future Work” section in the main paper.
>
> Q3 (Weaknesses part 3): "Bounds on modeling error"
>
> A3:
> As discussed in our response to Q1, we don’t assume that the bounds on the modeling error ‘d’ are known or provided by an oracle. Consequently, for all the experimental results shown in the paper, the bounds on ‘d’ were computed using a confidence interval, computed over the validation dataset constructed to evaluate the learnt model. To provide more information to the readers, we will move Table 1 from  Appendix Section A.4 to the main paper, which documents the modeling error values ‘d’ computed for the ablation study performed for the “safe land navigation” problem.
>
> Q4 (Minor comments part 1): "Model-based policy"
>
> A4:
> That is a good point which could have been made clearer in the main paper. Indeed, our method is agnostic to how the policy was developed, till we have a learned model of the dynamics. Often people build model and policy simultaneously, so we presented from that viewpoint, but we will clarify this fact throughout the paper. Thank you for bringing this point up.
>
> We will further incorporate your other comments in the manuscript.
>
> References:
>
> [3] Xiang, Tian, Fan Jiang, Qi Hao, and Wang Cong. "Adaptive flight control for quadrotor UAVs with dynamic inversion and neural networks." In 2016 IEEE International Conference on Multisensor Fusion and Integration for Intelligent Systems (MFI), pp. 174-179. IEEE, 2016.
>
> [4] Clarke, Shanelle G., and Inseok Hwang. "Deep Reinforcement Learning Control for Aerobatic Maneuvering of Agile Fixed-Wing Aircraft." In AIAA Scitech 2020 Forum, p. 0136. 2020.
>
> [5] Kim, Boo Min, Kwang Chan Choi, and Byoung Soo Kim. "Trajectory tracking controller design using neural networks for tiltrotor UAV." In AIAA Guidance, Navigation and Control Conference and Exhibit, p. 6460. 2007.
>
> [6] Kumar, Vikash, Emanuel Todorov, and Sergey Levine. "Optimal control with learned local models: Application to dexterous manipulation." In 2016 IEEE International Conference on Robotics and Automation (ICRA), pp. 378-383. IEEE, 2016.
>
> [7] Gu, Shixiang, Ethan Holly, Timothy Lillicrap, and Sergey Levine. "Deep reinforcement learning for robotic manipulation with asynchronous off-policy updates." In 2017 IEEE international conference on robotics and automation (ICRA), pp. 3389-3396. IEEE, 2017.
>
> [8] Gupta, Abhishek, Clemens Eppner, Sergey Levine, and Pieter Abbeel. "Learning dexterous manipulation for a soft robotic hand from human demonstrations." In 2016 IEEE/RSJ International Conference on Intelligent Robots and Systems (IROS), pp. 3786-3793. IEEE, 2016.
>
> [9] Chen, Mo, Jennifer C. Shih, and Claire J. Tomlin. "Multi-vehicle collision avoidance via Hamilton-Jacobi reachability and mixed integer programming." In 2016 IEEE 55th Conference on Decision and Control (CDC), pp. 1695-1700. IEEE, 2016.
>
> [10] Chen, Mo, Sylvia Herbert, and Claire J. Tomlin. "Exact and efficient Hamilton-Jacobi guaranteed safety analysis via system decomposition." In 2017 IEEE International Conference on Robotics and Automation (ICRA), pp. 87-92. IEEE, 2017.

---

> ### Author Response · Authors · 2020-11-24
> **Updated manuscript**
>
> We thank you for your comments.
>
> We have included a second paragraph in Section 5 - Conclusion, discussing the scalability issue of the proposed reachability algorithm and citing some ongoing works which are trying to address this issue (as mentioned in Q2).

---

### Decision · Program_Chairs · 2021-01-07
**Final Decision**

**Decision:**

Reject

**Comment:**

# Quality:
The paper makes a good job of presenting the proposed algorithm, which seems interesting and solid.
However, the paper fails to place the proposed approach in the larger context of the existing literature.
In addition, only qualitative results are presented, without any comparison.
As such, it is impossible to really understand the goodness of the proposed approach.

# Clarity:
The paper is generally well-written and clear.

# Originality:
The proposed approach is novel to the best of the reviewers and my knowledge.

# Significance of this work:
The paper deal with a very relevant and timely topic. However, as stated by the authors themself the paper is not concerned with high-dimensional systems, which is what would really differentiate this work compared to existing literature. In addition, the paper has no quantitative results nor comparisons against previous literature, and does not evaluate any of the standard benchmarks.

# Overall:
There is disagreement from the reviewers regarding the acceptance of this paper, and the overall score is very borderline. After thoroughly reading the paper, I agree with the evaluation of Reviewer 2 and 3 regarding the lack of comparisons and thus lean towards rejection.